# Integrin-Specific Stimuli-Responsive Nanomaterials for Cancer Theranostics

**DOI:** 10.3390/pharmaceutics16111441

**Published:** 2024-11-11

**Authors:** Zahra Taheri, Negin Mozafari, Ghazal Moradian, Denise Lovison, Ali Dehshahri, Rossella De Marco

**Affiliations:** 1Department of Pharmaceutics, School of Pharmacy, Shiraz University of Medical Sciences, Shiraz 71348-17336, Iran; zahra.taheri436shz@yahoo.com (Z.T.); mozafari.negin@gmail.com (N.M.); 2Student Research Committee, School of Pharmacy, Shiraz University of Medical Sciences, Shiraz 71348-17336, Iran; ghazalmoradian81@gmail.com; 3Department of Agricultural, Food, Environmental and Animal Sciences (Di4A), University of Udine, 33100 Udine, Italy; denise.lovison@uniud.it; 4Department of Pharmaceutical Biotechnology, School of Pharmacy, Shiraz University of Medical Sciences, Shiraz 71348-17336, Iran

**Keywords:** integrins, RGD, cancer, stimuli-responsive, nanoparticles, theranostic, diagnostic

## Abstract

**Background**: Cancer is one of the leading causes of death worldwide. The tumor microenvironment makes the tumor difficult to treat, favoring drug resistance and the formation of metastases, resulting in death. **Methods**: Stimuli-responsive nanoparticles have shown great capacity to be used as a powerful strategy for cancer treatment, diagnostic, as well as theranostic. Nanocarriers are not only able to respond to internal stimuli such as oxidative stress, weakly acidic pH, high temperature, and the high expression of particular enzymes, but also to external stimuli such as light and paramagnetic characteristics to be exploited. **Results**: In this work, stimulus-responsive nanocarriers functionalized with arginine-glycine-aspartic acid (Arg-Gly-Asp) sequence as well as mimetic sequences with the capability to recognize integrin receptors are analyzed. **Conclusions**: This review highlights the progress that has been made in the development of new nanocarriers, capable of responding to endogenous and exogenous stimuli essential to combat cancer.

## 1. Introduction

Cancer is a leading cause of death worldwide and is responsible for one in six deaths annually. An increase of 47% is estimated until 2040, equal to 28.4 million of new cancer cases compared to 19.3 million in 2020 [1]. The early diagnosis and treatment of cancer are prerequisite for health management. Conventional chemotherapy and radiotherapy lead to significant side effects due to a lack of selectivity, and surgery might be invasive for patients. The cardiovascular system, kidneys, liver, and lungs are the organs in which chemotherapy drugs and radiation induced by radiotherapy accumulate the most. This accumulation leads to serious side effects such as cardiotoxicity [2], nephrotoxicity [3], hepatotoxicity, and pulmonary fibrosis [4]. Most drugs have DNA in proliferating cells as a biological target. Consequently, tissues such as bone marrow and gastrointestinal tract, being rich in highly proliferating cells, lead to the formation of side effects such as gastrointestinal toxicity, myelosuppression, and immunosuppression [5]. Chemotherapy drugs and radiation also cause oxidative stress and inflammatory response because they present metabolic instability and can induce the formation of secondary tumors [6]. Hence, remarkable efforts are underway to increase the efficiency while maintaining the side effects as much as possible [7]. Immunotherapy and gene therapy arise a great deal of interest among the scientific community for the treatment of cancer despite their remarkable weaknesses including limited response rates, inability to predict clinical efficacy and potential side effects such as autoimmune reactions, while gene therapy is a high-priced resolution [8,9]. Amongst conventional treatments, metal complexes, particularly ruthenium complexes, are considered promising candidates as chemotherapeutic agents in addition to the platinum complexes (cisplatin, carboplatin, and oxaliplatin) [10,11,12]. The most important ones are Nami-A and KP1019 which were introduced into the clinical trials [13]. The most commonly used diagnostic are X-rays, endoscopy, ultrasound, magnetic resonance imaging (MRI), computed tomography (CT), positron emission tomography (PET), and single-photon emission tomography (SPECT). However, the low sensitivity and poor contrast between benign and malignant tissues render these techniques insufficient for early diagnosis. Theranostics is a novel strategy that combines therapy and diagnosis. This approach comprises the four principles of medicine or P4-medicine, which are prediction, prevention, personalization, and participation [14]. Nanomedicine is a defined branch of medicine that seeks to apply nanotechnologies in therapy and diagnosis. Nanomaterials are widely used in nanomedicine because of their small size, effective binding to the target, highly controllable physicochemical properties, and high delivery efficiency of the payload, as well as the potential to be complexed with multiple imaging modalities, and the potential combination of the payload with theranostic agents [15]. The main challenge for researchers is to develop stimuli-responsive nanomaterials capable of exploiting internal and/or external stimuli such as pH, reactive oxygen species (ROS), redox, enzymes present in the tumor environment, and external stimuli including light, proximity infrared (NIR), and magnetic field [16]. Stimuli-responsive nanomaterials have shown passive targeting characteristics in vascularized solid tumors due to the enhanced permeability and retention (EPR) effect [17]. However, recent studies have shown that EPR is not the only way for non-targeted nanomaterials to accumulate in the tumor sites. The complexity of tumor microenvironment and patient-to-patient variations in terms of gender, age, and other diseases [18] have led researchers to seek novel alternative strategies in order to transfer the payloads into their precise site of action. One of these successfully tested approaches is targeting the tumor microenvironment using ligand-mediated strategies [19]. In order to improve the ability of the nanomedicine to recognize the tumor sites, ligands capable of selectively binding cancer cells are preferably used. It has been demonstrated that the integrin ligand is over-expressed in tumor blood vessels and in various invasive tumor cells, while its expression on normal cells is not considerable. Integrin ligand-conjugated nanomaterials are internalized via clathrin or caveolin-1 internalization pathways rapidly. Since efficient cancer treatment might be hampered due to different resistance mechanisms, the receptor-mediated endocytosis leads to the intracellular delivery of payloads resulting in efficient therapeutic effects. In addition, there have been several reports on the application of integrin ligands as targeting moieties to transfer various types of delivery systems to the integrin-overexpressed cells. The effects of integrin receptors in tumor growth, metastasis, and angiogenesis have attracted great attention not only to use them as a target for cancer therapy but transferring various payloads into the cells over-expressing these receptors. The integrin-targeted nanomaterials may have several advantages including (i) the reduction of non-specific interactions with normal cells, tissues, and organs which subsequently leads to lower side-effects, (ii) the accumulation of payloads in the precise site of action, higher concentrations of the drug in the target cell or tissue and subsequently the reduction of administrated primary dose, (iii) the reduction of chemo-resistance due to the opportunity to deliver other therapeutic agents with the capability to overcome multi-drug resistance. Thus, integrin is a promising target for the tumor treatment and diagnosis [20]. In this review, we aim to discuss integrin stimuli-responsive nanomaterials in theranostics and the future of this modality for treatment and diagnosis.

## 2. Transmembrane Receptors

Transmembrane receptors are a large family of proteins formed by more than 800 receptors and 35 heterotrimeric G proteins subunits [21], which cross the cell membrane seven times and thereby are also called seven trans-membranes (7TM). They are characterized by an α-helical structure with an extracellular amino-terminus and an intracellular carboxy-terminus. The term 7TM receptors refers to G-protein-coupled receptors (GPCRs) because signal activation involves the heterodimeric G-proteins [22]. GPCRs were classified on the basis of their sequence homology and functional similarity into six classes: family A or rhodopsin-like receptors, family B or secretin receptor, family C or metabotropic glutamate receptors, family D or parasitic mating pheromone receptors, family E or cyclic AMP receptors, and family F or frizzled and smoothened receptors [23]. Human GPCRs on the basis of the phylogenetic studies are classified into five classes and called “GRAFS”: Glutamate (G), Rhodopsin (R), Adhesion (A), Frizzled (F), and Secretin (S) [24]. The different classes are summarized in Table 1.

The fundamental role of GPCRs is to transduce the extracellular stimuli to intracellular signals [37]. Activation of GPCRs is accomplished via ligand-binding. GPCRs are coupled to heterotrimeric G proteins that are formed by three subunits: Gα, β, and γ. Gα subunits are themselves a member of Gi/o, Gs, Gq/11, and G12/13. The interaction of each subunit with a specific target leads to the modulation of t cellular activity [38]. Once GPCRs interact with the heterotrimeric G, the subunit Gα dissociates from the β and γ subunits and consequently interacts with the β subunit of the integrin, leading to the activation of signaling pathways inside and outside the cell [39].

## 3. Integrins

Integrins are heterodimeric transmembrane receptors and are formed by two subunits α and β, their combination results in the generation of 24 different receptors with different properties [40]. The two subunits form dimers via non-covalent bonds. The dimer is formed by an NH_2_-terminal extracellular domain, transmembrane domain, and cytoplasmic tail, and is located across the cell membrane (Figure 1).

The extracellular domain is situated on the cellular surface and binds and interacts with the ligands of the extracellular matrix, such as fibronectin, vitronectin, collagen, and various adhesion receptors. The extracellular domain is linked to a cytoplasmic tail by the transmembrane domain which crosses the cell membrane. The cytoplasmic tail acts as an adaptor protein for intracellular signaling in the formation of focal adhesions and interactions with the cytoskeleton. Integrins are involved in several processes such as cell-cell adhesion and cell-extracellular matrix adhesion, migration/invasion, proliferation, survival, and apoptosis [41,42]. Furthermore, integrins are responsible for activating inside-out and outside-in signaling pathways which are crucial for the exchange of information between ECM proteins and intracellular molecules [39]. Most of the integrins interact with the Arginine-Glycine-Aspartic Acid (RGD) sequence in the presence of the divalent cations, the so-called metal-ion-dependent adhesion site (MIDAS) motif [43]. The RGD sequence was first defined by Ruoslahti et al. in 1984 as the highest conserved fibronectin conserved sequence [44]. In such sequence, arginine forms a salt bridge with the aspartate within the β-propeller while the carboxylic group of the aspartate forms polar interactions in the βA domain. Glycine acts as a spacer and forms hydrophobic interactions only with αv subunit [43]. Kessler et al. introduced a tripeptide sequence into the cyclic pentapeptide c[RGDfK] to improve the metabolic stability and enhance the bioavailability. Kessler and colleagues observed that the phenylalanine exhibiting D stereochemistry nearby the aspartic acid leads to the correct conformation for interacting with the receptor. Importantly, the role of *N*-Me-Val was not crucial for the activity of the peptide, allowing its replacement by other amino acids [45]. The replacement of *N*-Me-Val with Lys in the cyclic structure provides an opportunity to conjugate the pentapeptide sequence to nanoparticles for diagnostic and therapeutic approaches [46,47,48].

## 4. Stimuli in Cancer Therapy

Despite outstanding progress in cancer treatment, cancer still kills millions of people worldwide [49]. Significant angiogenesis, diverse metabolic pathways, and rapid proliferation, as well as high oxidative stress, high expression of some enzymes, and slight acidic conditions make the tumor microenvironment very different compared to other parts of the tumor. All of these factors lead to a poor response of tumor tissues towards the treatment and progression of metastasis [50]. Due to the unique properties of the tumor microenvironment, such as lymphatic drainage and enhanced permeability and retention effect (EPR), nanocarriers are promising vehicles for cancer-targeted drug delivery systems [51]. Although they have been used to deliver therapeutic or imaging agents to tumor cells, several barriers must be overcome for their wide clinical applications. These barriers are included but not limited to burst release, degradation, and leakage of cargos [52].

Stimuli-responsive nanoparticles (SRNs) have been developed to control the rate, location, and time of drug release. Recently, SRNs have attracted great attention for the cancer theranostics and treatments [49,53]. SRNs are materials with specific ability to be triggered by signals from out- or inside the body for precision activation or release (Figure 2) [54].

These systems are functionalized to deliver their cargos in specific sites through responding to external or internal stimuli [55]. The mechanism of action of SRNs can be explained as follows: they are administered and accumulated in tumor tissues through active or passive targeting. Subsequently, the system is activated due to the specific stimuli and releases the cargo [56]. The trigger or stimulus is a force that initiates a cascade of events leading to the nanoparticle activation for diagnosis and/or treatment. The stimulus can be part of the tumor tissue or its source can come from outside the body. SRNs have recently attracted attention because of their special ability for controlled drug release [57]. These stimuli cause charge conversion, drug activation, ligand exposure, size or structure conformation, or even change the biological activity of nanocarriers and promote controlled drug release. SRNs can overcome multi-drug resistance for the treatment of cancer [52]. In addition, they provide efficient concentration of the therapeutic agent in the tumor microenvironment/tumor cells [58]. Considering the pathological conditions of the tumor microenvironment, tumor tissue, and intracellular compartments, different SRNs can be designed, with the aim of increasing biological activity, efficiency, specificity, and even overcoming multi-drug resistance. SRNs can also include nanoparticles loaded with a pro-drug, which are activated in response to tumor stimuli for precision therapy [52]. Several stimuli-responsive nanocarriers have been reported in clinical trials. Liposomes are considered promising and versatile drug vesicles and used to designed a smart liposomal system able to respond to internal or external stimuli [59]. They are formed by natural phospholipids that mimic the properties of biological membranes. By incorporating a polyethylene glycol (PEG) coating, they have revealed a long circulation. Liposomal formulations are able to improve the pharmacokinetics and pharmacodynamics of the drug, reducing its toxicity [60]. Finally, the small size promotes the permeation and retention (EPR) effect, allowing passage into blood vessels and accumulation in tumor tissues [61]. An interesting approach in clinical trials is liposomal cisplatin formulation (LiPlaCis), a drug delivery system formed by liposomes nanocarriers loaded with cisplatin. The system is developed to be degraded by secretory phospholipase A_2_ (PLA_2_), an enzyme presents in high concentrations in different cancer types and is used for triggering drug release into the organs/tissues. LiPLaCis shows greater distribution and reduced side effects when compared to the free drug [62]. Among the developed nanoparticles, magnetic nanoparticles are of great interest. In the presence of an alternative magnetic field, they generate local hyperthermia to trigger drug release and tumor ablation. The interaction between magnetic nanocarriers and magnetic field facilitates magnetically driven accumulation of nanocarriers in tumors [52]. Biological tissues are transparent to magnetic fields, allowing magnetic targeting to be followed remotely and non-evasively. With this system, magnetic targeting is more flexible and does not depend on specific receptor expression [63]. Iron oxide (MION) and maghemite (gFe_2_O_3_) nanoparticles exhibit low toxicity and high biocompatibility, and thermal effects under different applied magnetic fields and small size are in clinical trials and show interesting results for the treatment of hepatocellular carcinoma and prostate cancer [52]. Another interesting approach is to use polymeric micelles which exhibit a robust core-shell structure, kinetic stability, and ability to solubilize hydrophobic drugs. Such nanoparticles are able to release the drug in the acidic environment of the endosomal or lysosomal compartments of the target tumor cell, using the different pH present in the normal cells [64]. The use of nanoparticles for cancer treatment is increasing, but nanomaterials have an inherent toxicity that can trigger the accumulation of metals into the tissue, side effects, and drug release at a wrong site [65]. The need to develop innovative strategies to reduce nanotoxicity is the challenge that must be addressed to use nanomaterials safely [66]. Some of these systems are listed in Table 2.

### 4.1. Endogenus Stimuli

#### 4.1.1. pH-Responsive Nanocarriers

Malignant tumors are characterized by low extracellular and interstitial pH, where metabolites, such as lactic acid and CO_2_, result in a slightly acidic environment [67,68]. In most of cancer cells, aerobic glycolysis pathway is the main way to generate energy, which can result in the production of large amount of lactic acid, in contrast to the oxidative phosphorylation pathway in normal cells [69,70,71]. The pH value in normal cells is approximately 7.4 higher than in tumor tissues (~6.5). The above pH gradient could be advantageous as an endogenous stimulus for stimuli-responsive nanoparticles [72,73]. Acidic extracellular pH plays an important role in the enhancement of cancer therapy [71]. There are different strategies for the synthesis of pH-responsive nanoparticles. The first strategy involves the use of peculiar molecular structures with specific functional groups in the nanoparticles backbone that can be protonated under acidic conditions. These phenomena cause the destruction of the hydrophilic–hydrophobic equilibrium and the destruction of the nanoparticles, leading to drug release by rearranging, expanding, or disintegrating the chemical structure [74]. The second strategy involves the role of chemical bonds, which are stable at normal pH and sensitive to lower pH values. This is known as cleavable acid-responsive bonds. In this approach, nanoparticles would be able to release the drug due to the cleavage of chemical bonds to form the nanostructure [73,75]. The use of pH-Low Insertion Peptides (pHLIPs) is the last strategy. pHLIPs are 36-aa peptides with a high affinity for the cell membrane at physiological pH. At lower pH values, the protonation of Asp and Glu enhanced the hydrophobicity of the peptide. This can affect peptide folding and the consequent insertion into the cell membrane, improving its cellular uptake [71,76]. In one study, Zhang et al. worked on pH-responsive peptide-modified polymeric micelles, containing Paclitaxel (PTX). They designed a H_7_K(R_2_)_2_ with pH-triggered cell-penetration activity. These peptides were conjugated to PLGA-PEG. H_7_K(R_2_)_2_-modified polymeric micelles containing PTX were prepared and characterized. The in vitro release of PTX was observed at pH 6.8 by flow cytometry and confocal microscopy, leading to the inhibition of the growth of human breast adenocarcinoma cells [77]. In another study, they tried to design pH-responsive system containing Doxorubicin (DOX) for enhancing glioma targeting and anti-tumor activity. In this study, H_7_K(R_2_)_2_-modified pH-sensitive liposomes containing DOX were investigated. Liposomes show high sensitivity to the acidic environments and are destabilized under these conditions, enhancing cellular uptake with rapid intracellular release. In vitro tests were conducted on C6 (rat C6 glioma cells) and U87-MG (human glioblastoma cells), and a higher cytotoxic effect was observed compared to normal cells. These data were also confirmed by in vivo tests conducted on nude mice bearing orthotopic GBM U87-MG or C6 tumor models [78]. In 2019, Motealleh et al. developed a 3D-nanocomposite (NC) hydrogel system for pH-responsive surface-mediated delivery of Doxorubicin. The 3D-NC hydrogel was formed using periodic mesoporous organosilicate (PMO) nanoparticles loaded with DOX, where the outer shell was functionalized with poly-L-lysine (PLL) and subsequently embedded into the alginate hydrogel network. PLL is responsible for the electrostatic interactions with the cell membrane and its cellular uptake. The 3D-NC hydrogel reduced the viability of Colo 818 cancer cells owing to the high drug release in the local acidic environment of the Colo 818 compared to normal fibroblast cells [79]. These examples demonstrate promising results regarding the anticancer activity of pH-responsive nanocarriers.

#### 4.1.2. Enzyme-Responsive Nanocarriers

Another strategy for designing drug delivery systems for cancer therapy is the adoption of enzyme-responsive nanocarriers. In the human body, biological activity and metabolic processes are carried out by enzymes with high specificity for their substrates [80]. In contrast to normal cells, tumor cells overexpress specific enzymes such as proteases (matrix metalloproteinases (MMPs) and cathepsin-B), phospholipases, and glycosidases. According to the said literature, stimuli-responsive nanocarriers contain substrates that can be cleaved by particular enzymes present in tumor cells and consequently release the desired cargo at tumor sites [81,82]. MMPs degrade proteins by their proteolytic activity. Compared with normal cells with inhibited MMPs activity, tumor cells overexpress some MMPs, including MMP2, MMP9, and MMP13 [81,83]. Among these enzymes, MMP2 play a critical role in tumor metastasis and invasion [84]. These overexpressed MMPs have been widely exploited by scientists. For example, Zhu et al. attempted to enhance the anticancer activity of PTX by designing a PEG2000-Peptide-PTX conjugate, which responds to upregulated MMP2. The results of the in vitro and in vivo tests revealed the great potential of cancer cell-selective intracellular delivery of an anticancer drug for enhanced cancer chemotherapy [85]. In another study, Liu et al. constructed a drug delivery system based on mesoporous silica nanoparticles (MSN). They used PLGLAR as a peptide that is sensitive to MMP13, bovine serum albumin as an endcap for sealing MSNs, lactobionic acid as a targeting ligand, and DOX hydrochloride as a model drug. These results demonstrated the ability of the system to efficiently inhibit tumor growth with minimal side effects [86]. In another study, Kalafatovic et al. designed a self-assembled micelle system for loading DOX, which was sensitive to MMP9 [87]. Cathepsins are lysosomal cysteine proteases that are either expressed on the cell surface or are released into the extracellular matrix in pathological conditions [88]. Compared to normal tissues, cathepsin B (Cat-B) has a higher concentration and activity in cancer cells. This enzyme can hydrolyze various sequences including Arg-Arg, Ala-Leu, Phe-Arg, Phe-Lys, Ala-Phe-Lys, and Gly-Phe-Leu-Gly [81]. For instance, the Gly-Phe-Leu-Gly sequence has been linked to hydroxypropyl methacrylate (HPMA) copolymer conjugates to release the drugs into lysosomes by the cleavage of the peptide spacer and Cat-B [89]. Recently, Gly-Phe-Leu-Gly oligopeptides have been adopted as enzyme-responsive cores, and DOX as a model anticancer drug. This nanoparticle is degraded by Cat-B and releases the drug efficiently [90].

Phospholipases hydrolyze phospholipids into lipophilic substances and fatty acids. One of the mostly investigated phospholipase family is phospholipase A2 (PLA2) which is divided into intracellular and secretory PLA2 [91,92]. Lee et al. designed a strategy for imaging of prostate cancer cells by using biocompatible up-conversion nanoparticles (UCNPs)-loaded phosphate micelle that could be cleaved by secretory PLA2 [93]. In another study, Ghavami et al. utilized phospholipase sensitive liposomes (PSLs) to encapsulate antisense octaarginine peptide nucleic acid conjugate, which responds to secretory PLA2 [94].

Glycosidases are another family of enzymes that trigger release of drug to malignant sites. They can hydrolyze carbohydrates in lysosomes and play a key role in N-linked glycosylation in the endoplasmic reticulum (ER) and Golgi apparatus [95]. Clahaut et al. worked for the first time on doxorubicin-folate conjugate, which is sensitive to β-galactosidase to induce cell death in acute myeloid leukemia blasts. These folate positive cells efficiently responded to the enzyme-responsive nanocarrier with low adverse effects [96]. In another study, Rastegari et al. designed two different magnetic nanoparticles with β-cyclodextrin and chitosan, loaded with prodigiosin as a model anticancer drug. The results of the study showed that the drug was released by hydrolytic lysosomal enzymes such as α-glucosidase [97]. Kolodych et al. developed a new class of β-galactosidase-cleavable linkers for antibody-drug conjugates (ADCs) by employing trastuzumab as a model drug and loading it using galactoside linkers, monomethyl auristatin E (MMAE), and cysteine-reactive groups. The in vivo results demonstrated that ADCs with galactoside linker have superior therapeutic efficacy than trastuzumab alone [98].

#### 4.1.3. Redox-Responsive Nanocarriers

Reactive Oxygen Species (ROS) include the hydroxyl radicals (HO•), the superoxide anions (O_2_^−^), hydrogen peroxide (H_2_O_2_), and singlet oxygen (^1^O_2_) which are usually produced by incomplete reduction of oxygen [99]. Compared to normal cells, the ROS concentration is much higher in cancer cells, which can be used for preparing drug delivery systems [100]. For example, Wilson et al. constructed thioketal nanoparticles (TKNs) using poly- (1,4-phenyleneacetone dimethylene thioketal) that can deliver small interfering RNA (siRNA). TKNs can be degraded at high ROS concentration and release the siRNA at the site of action [101]. In addition, Lux et al. designed a system containing boronic esters that can be degraded in the presence of H_2_O_2_ and release cargos [102]. Recently, Du et al. prepared thioether phosphatidylcholines (S-PCs) and S-PC-based liposomes (S-LPs) loaded with DOX as a model anticancer drug. The results demonstrated higher efficiency of DOX/S-LPs resulting from the rapid release of drug from the conjugate by excess amount of ROS [103].

Glutathione (GSH) is a tri-peptide that contains glutamate, cysteine, and glycine (L-γ-glutamyl-L-cysteinyl-glycine). Despite its crucial role in detoxification, antioxidant activity, and cell cycle regulation, high GSH concentrations result in different pathological conditions, including cancer and neurodegenerative diseases. GSH levels in cytoplasm are approximately 1–10 mM, which is higher than that in the extracellular environment (2–20 µM) [104]. GSH concentration in tumor tissues is at least four-fold higher than that in normal cells. The differences in GSH levels were exploited to develop novel drug delivery systems [105,106,107,108]. Therefore, chemical groups, which are sensitive to high level of GSH including disulfide and diselenide bonds, as well as manganese dioxide (MnO_2_) have been used in recent studies [109,110,111]. For instance, Li et al. designed a novel drug carrier modified with disulfide-bound PEG and amide-bound polyethylenimine (PEI). High GSH level results in di-sulfide degradation and drug release. This nanocarrier improves the tumor accumulation capability and decreases the side effects of chemotherapy [112]. Recently, Chai et al. constructed a novel drug delivery system by conjugating ibuprofen (BF) to the hyaluronic acid (HA) backbone through a disulfide bond (HA-ss-BF), which can self-assemble into micelles for DOX delivery. This drug delivery system is efficient because of its ability to target cancer cells by recognition of CD44 receptors, inhibition of overexpressed cyclooxygenase-2 in cancer cells, improvement of cellular uptake, and improved biodistribution [113]. Additionally, selenium has special properties, including high reactivity and sensitivity owing to its low electronegativity and high atomic radius. In addition, the Se-Se bond has a lower bond energy that the S-S bond resulting in faster cleavage and faster drug release [114,115,116]. In this context, Wei et al. designed a nanocarrier based on amphiphilic polyester urethane with multiple Se-Se bonds which were loaded with DOX. According to these results, the antitumor activity of the DOX was enhanced in this system [116]. Recently, MnO_2_ has attracted increasing attention as a GSH-responsive trigger. For example, Zhang et al. constructed a drug delivery system for chemotherapy that reduced the side effects of Bleomycin (BLM). In this study, PEG-modified mesoporous manganese dioxide (HMnO_2_@PEG) nanoparticles were used to load the antitumor drug BLM. At the tumor site, degradation of this system and drug release occurred simultaneously because of an excess amount of GSH. Then, BLM-Mn^2+^ formation led to the therapeutic activity of BML while maintaining the side effects as low as possible [117].

#### 4.1.4. Temperature-Responsive Nanocarriers

High temperatures are another major trigger for the design of stimuli-responsive drug delivery systems. Temperature can be employed as an internal stimulus due to the pathological conditions including the hyperthermic nature of the tumor or inflammation. Additionally, temperature can be used as an external stimulus by utilizing external heat sources [118,119]. Therefore, nanocarrier should be stable at body temperature and release the drug when tumor temperature raises. Thermoresponsive polymers exhibit phase-transition, which defines their solubility and Critical Solution Temperature (CST). If the temperature is greater than the lower critical solubility temperature (LCST) of the system, dehydration of the polymer occurs. Consequently, the hydrophobic polymer collapses and releases the drug [120,121,122,123]. Poly-N-isopropylacrylamide (PNIPPAm) is a polymer that is widely used because of its LCST of approximately 35 °C [124,125]. In one study, Xu et al. synthesized a drug carrier using diselenide-bond-linked poly-N-isopropylacrylamide which was then loaded with PTX. This conjugate can self-assemble into core-shell micelles that are simultaneously temperature- and redox-responsive. These results demonstrated that the conjugates are safe and show promising anticancer efficacy [125]. Besides thermoresponsive polymers, thermoresponsive liposomes have been widely investigated for smart drug delivery applications [126]. For example, Thanou et al. loaded topotecan in thermo-responsive liposomes and improved the drug uptake into cancer cells [126]. In another study, Ta et al. prepared a polymer-modified thermo-sensitive liposome (pTSL) loaded with DOX. This system exhibits dual pH/temperature-dependent phase transition properties owing to the coexistence of temperature-responsive N-isopropylacrylamide and pH-responsive propylacrylic acid. By attaching to liposomes, these copolymers were membrane-disruptive in a pH/temperature-dependent manner. pTSL improves release profile and is stable in serum with minimal drug leakage [127].

### 4.2. Exogenus Stimuli

#### 4.2.1. Light-Responsive Nanocarriers

Nanocarrier uptake and tumor penetration are obstacles that result from the high interstitial pressure and dense extracellular matrix of the tumors. An invasive strategy giving the solution to these problems, which attract considerable attention, is photo-thermal therapy (PTT). Most photoactive chromophores are sensitive to ultraviolet (UV) radiation. Nevertheless, UV light has two disadvantages including poor penetration depth in human tissues and its high phototoxicity and carcinogenic effect on normal cells and tissues [128]. Compared to UV light, near-infrared (NIR) light is more attractive because of its deeper penetration rate and lower cell toxicity. Hence, PTT is induced by light, preferably NIR, which relies on the absorbance ability and the transformation of NIR energy to heat [129,130,131,132]. Besides PTT, there is another strategy for utilizing lights termed photodynamic therapy (PDT). PDT is a minimally invasive method with low side effects that induces cell death via light irradiation of a photosensitizer and generation of highly reactive singlet oxygen species [133,134]. In one study, Chen et al. utilized indocyanine green (ICG), a widely used NIR dye in photothermal therapy, and DOX as a model anticancer drug loaded on a nanoparticle of arylboronic ester and cholesterol modified hyaluronic acid. This combination demonstrated a photothermal effect and a faster trigger release of DOX with NIR irradiation [135]. An interesting approach is proposed by Diaz and co-workers. They used the laser-induced thermal mechanism to give de-hybridization of DNA or RNA sequences from gold nanoparticles. The nanoparticles are loaded with complex hybridizing molecules: a single-stranded complementary DNA, a DNA duplicator, and a basically modified DNA duplicator. DNA duplicators show a different melting point, and when light hits the nanoparticles, causing an increase in temperature, it allows the release of the labeled DNA duplicator in the cancer cells. The system offers the possibility of greater control of drug release [136]. Chen et al. proposed to use red-light-responsive metallopolymer nanocarriers named PolyRuCHL formed by hydrophilic poly(ethylene glycol) (PEG) block and a hydrophobic ruthenium (Ru)-containing block as red-light-responsive linked with chlorambucil (CHL). PolyRuCHL is loaded with DOX. The advantage of red light is the ability to penetrate deeply into tissue for PTT. Red-light irradiation induced a cleavage of Ru-CHL triggering DOX release from the nanoparticles. The authors observed a synergic effect to inhibit the growth and multidrug resistance in breast cancer cell line MCF-7R [137]. The approach used by Fan and co-workers draws inspiration from the properties of liquid metals (LMs) that exhibit excellent photothermal conversion efficiency, generating heat under NIR laser irradiation. To prepare LMs, they used a gallium–indium eutectic alloys (EgaIn) which shows excellent combination of thermal-conductivity, transformability, and a high biocompatibility. The authors prepared a poly (NIPAm-co-MBA) hydrogel (PNM) contained with LM droplets and encapsulated it with DOX to form the final system PNM/LM/DOX. After NIR irradiation, the temperature of the system rises above the lower critical temperature of the solution, which causes the hydrogel to change shape and size. The hydrogel shrinks, inducing a simultaneous release of the aqueous solution and DOX. This controlled release can reduce the amount of drug into normal tissues and thus the side effects [138].

#### 4.2.2. Magneto-Responsive Nanocarriers

Magneto-responsive nanocarrier design is a novel strategy for the development of drug delivery systems. This method has multiple advantages including the following: (1) Superparamagnetic iron oxide nanoparticles act as a contrast agent in magnetic resonance imaging (MRI), which means that they are suitable for drug delivery and as diagnostic imaging agents simultaneously [139], (2) Compared to UV light, magnetism has not shown any particular side effects in the human body [140], and (3) It has no physical interactions with the human body [122]. Although it has several advantages, its low accuracy can be a barrier to its clinical translation [122]. In a study, Yoon et al. designed a dual targeting nanoparticle, which was sensitive to GSH and magnetism. In this work, methoxy PEG was grafted to water-soluble chitosan, DOX conjugated to the backbone of chitosan via disulfide linkage (which made it GSH sensitive), and finally, the conjugation of iron oxide made it magnetically responsive. The in vivo test demonstrated that this system has promising anticancer drug-targeting properties [141]. In Table 3, we summarized a number of nanomaterials and their components that can respond to different stimuli.

## 5. Stimuli-Responsive Nanomaterials Targeting Integrins

To achieve therapeutic and diagnostic goals, stimuli-responsive nanomaterials can also be designed for theranostics (Figure 3) [179,180].

This modality provides great opportunity for personalized cancer treatment [181]. Theranostic systems provide early specific personalized diagnosis and treatment, longer blood circulation and high resolution for imaging reagents, accumulation of loaded cargo in the tumor tissue, and controlled drug release, which together improve the efficiency and selectivity of treatment and the accuracy of tumor diagnosis with minimum side effects [49,181]. A theranostic system must have optimized diagnostic and therapeutic properties, cost-benefit of efficacy versus toxicity, improved pharmacokinetic and biocompatibility properties, and tumor-targeted ability [182,183]. In Table 4, some examples of stimuli-responsive nanocarriers targeting integrin receptors in cancer therapy are summarized.

### 5.1. Light-Responsive Theranostic Nanomaterials

Sheng et al. designed biomimetic HDL-like theranostic nanomaterials for personalized cancer therapy as part of a successful investigation of light-responsive theranostic nanomaterials for integrin receptors. For this purpose, the plasma was extracted and purified to extract Apo A. iRGD was conjugated on the Apo A surface by the Sulfosuccinimidyl 4-(nmaleimidomethyl) cyclohexane-1-carboxylate (sulfo-SMCC) as a cross-linker and indocyanine green was loaded as a diagnostic. The hydrodynamic size of nanoparticles was 86.7 ± 1.4 nm, which is desirable for passive targeting to tumor site via the EPR effect. The stability of the nanoparticles was evaluated in PBS for 4 weeks and in fetal bovine serum for 12 h, revealing that the formulation was stable, without precipitation or aggregation. The prepared nanoparticles could accumulate in tumor sites due to recognition by SR-BI receptors of tumor cells and the EPR effect, while they are able to bind to tumor cells and vessels specifically through iRGD interaction with αvβ3 receptors. The targeting was evaluated in 4T1 cells overexpressing of αvβ3. Confocal Laser Scanning Microscopy (CLSM) showed the strongest signal from the treated cells with nanoparticles compared to nanoparticles without iRGD or indocyanine green alone. Therefore, the conjugation of iRGD provided higher targeting capability and tumor homing ability of the nanoparticles. The prepared nanoparticles showed controlled drug release in the blood, and burst release upon NIR irradiation. Therefore, the iRGD conjugated HDL-like nanoparticles were capable to improve phototherapeutic and photodynamic effects [194].

### 5.2. Enzyme-Responsive Theranostic Nanomaterials

A promising investigation conducted by Cheng et al. developed an integrin-targeted delivery system displaying aggregation-induced emission (AIE) as a theranostic system. The prepared nanoparticles were conjugated by DGR for integrin-targeting, KRRRR peptide for nucleus localization, RRRR as a cell-penetrating peptide and endosomal escape, and AIE as a fluorescent probe for cell imaging, which were assembled with an antisense single-stranded DNA oligonucleotide. The self-assembly ability of peptides and DNA is based on electrostatic interactions, leading to a reduction in the non-specific adsorption and cationic toxicity. The prepared nanoparticles successfully delivered the antisense into the nucleus and showed a remarkable tumor-suppressive effect. In vivo experiments in tumor-bearing mice showed that the prepared nanoparticles accumulated in the tumor, kidney, and liver after 12 h. In addition, the injected nanoparticles diffused quickly into the tumor tissue within an hour. The decreased tumor volume in MDA-MB-231 tumor-bearing mice receiving theranostic nanoparticles, compared to the increased tumor volume in the control group, confirmed the efficiency of the prepared nanoparticles [195]. Another enzyme-responsive theranostic nanoparticle was developed by Hu et al. [196]. They designed mesoporous silica functionalized with MMP-2-responsive fluorescence imaging peptides and cRGD. The delivery system was loaded with camptothecin. The mesoporous silica nanoparticles showed a hexagonal and uniform mesostructure in X-ray diffraction with an average diameter of 150 nm. The targeting effect of cRGD was evaluated in overexpressing cRGD cell line (SCC-7) and low-expressed cRGD cell line (COS7). The results revealed that mesoporous silica nanoparticles had a higher concentration in SCC-7 cells than in mesoporous silica nanoparticles without the cRGD ligand. High amounts of MMP-2 in tumor tissues cleaved fluorescence imaging peptides on the surface of nanoparticles, leading to both tumor imaging and the controlled release of camptothecin from the pores of mesoporous silica nanoparticles. Drug release determination showed that more than 87% of camptothecin was released in 36 h, whereas this amount was 35% in the absence of MMP-2. The results showed a promising approach for integrin-mediated enzyme-responsive theranostics.

### 5.3. pH-Responsive Theranostic Nanomaterials

To demonstrate the ability of pH-responsive theranostics for integrin receptors, the following investigations are examples of success in this field. In a study, perfluorocarbon nanoparticles encapsulated with a prodrug to target the transcription factor c-MYC (MI3-PD) were developed by Weilbaecher et al. Their findings were based on testing the above-mentioned nanoparticles in a murine breast cancer cell line (PyMT-Bo1, MFI 17), human melanoma cell line (MDA.MB.435, MFI 27), and human endothelial cell line (HUVEC, MFI 42) in vitro. The results showed that the number of M2 macrophages was reduced in 4T1 breast cancer cells treated with αvβ3-MI3-PD-NP. It was observed that the uptake of αvβ3-NP labeled with rhodamine was dependent on surface integrin expression. CD11b-expressing myeloid cells in the bone marrow were used for in vivo testing. Down-regulation of CD206 and arginase 1 in M2 macrophages suggests that nanoparticle-mediated drug delivery of MI3 MYC inhibitors could reduce M2 macrophage polarization and function in vivo [197]. Sun et al. developed iron oxide nanoparticles in the range of 2.5–5 nm. They proposed a new method for MRI detection of tumors by using Fe_3_O_4_ NP in vivo, introducing ultrasmall Fe_3_O_4_ NPs (<10 nm in hydrodynamic diameter) that have specific targeting ability for tumors. Small hydrodynamic sizes are known to be capable of specific uptake and extravascular ability. To prepare these NPs, they used the thermal decomposition of Fe(CO)_5_ and air oxidation combined with a different strategy (using the 4-methylcatechol (4a-MC)). They showed that Fe_3_O_4_ NPs were mainly accumulated in integrin-overexpressing tumor vasculature and tumor cells. Moreover, they showed that these NPs did not accumulate in the macrophages [198]. In another study, Liao et al. developed flaky black phosphorus (BP) nanosheets with a lateral size of approximately 200 nm and a thickness of 5.46 ± 1.48 nm. In these nanosheets, BP causes pH sensitivity. The measurement of Nrf2/HO-1 is one of the most important key factors for oxidative stress mechanism in cells. The researchers prepared BP-Dox and observed an increased Dox release under acidic conditions in vitro. BPs showed photothermal effects, and after laser irradiation at 808 nm, Dox release was accelerated. However, BPs induce a mild inflammation in healthy mice and oxidative stress in the liver and lung, leading to cell apoptosis. To understand the biological effects of BPs, researchers prepared red blood cell (RBC) membrane-coated BP-conjugated RGD peptides (RGD-RBC@BP) via lipid insertion and co-ultrasound methods for effective photothermal therapy (PTT) of cancer through a targeted strategy. In vitro tests on HeLa cells showed increased cellular uptake, biocompatibility, and photothermal properties. Animal studies showed a prolonged circulation time and accumulation in tumor cells over expressing αvβ3 integrin [199].

### 5.4. Temperature-Responsive Theranostic Nanomaterials

Another successful strategy for the integrin-mediated delivery of theranostics is the application of temperature-stimuli systems. For instance, Lei et al. developed a temperature-responsive theranostic mesoporous silica nanoparticles loaded with indocyanine green (ICG) and doxorubicin. The system was decorated with RGD to target tumor tissues, as well as PEG, to protect mesoporous silica nanoparticles in the blood circulation, and an MMP-sensitive substrate (PLGVR) to remove PEG in tumor tissues through a host-guest interaction [196]. TEM images confirmed the successful synthesis of mesoporous silica nanoparticles with an average diameter of 60 nm. Both doxorubicin and ICG were loaded by a thermally cleavable gatekeeper that was cleaved due to hyperthermia induced by ICG under NIR illumination. For the in vivo evaluation of tumor imaging and targeting, mesoporous silica nanoparticles were injected intravenously into 4T1 tumor-bearing mice. The results showed that the near-infrared fluorescence (NIRF) of ICG in mice treated with mesoporous silica nanoparticles was detected at 8 h in tumor tissues with high potency compared to other tissues. Significant NIRF was detected at 24 and 48 h indicating the tumor-targeting ability and long-term tumor retention of the nanoparticles. Fluorescence imaging of frozen tumor sections showed weak fluorescence of doxorubicin in mice treated with ICG and doxorubicin, indicating poor retention ability and tumor targeting of free doxorubicin. The intensity was also low in tumor tissues of mice treated with mesoporous silica nanoparticles without NIR irradiation. The red fluorescence intensity of doxorubicin in mice treated with mesoporous silica nanoparticles and NIR was much stronger confirming the targeted drug delivery of doxorubicin with efficient drug release. In other words, released ICG made a light-to-heat conversion, and provided both imaging and burst drug release in the cytoplasm. In another study, RGD and ICG-decorated albumin nanoparticles loaded with artemisinin were synthesized. TEM images showed uniform spherical shapes with an average diameter of 141 nm and a zeta potential of –20 mV. Albumin nanoparticles have shown promising biocompatibility, stability, and temperature responsiveness. The particle size of the nanoparticles increased when exposed to high temperatures generated by ICG under NIR irradiation, leading to the drug release. The drug release study revealed that 61.5% of artemisinin was released after six cycles of NIR irradiation, which proved that the drug had a burst release upon NIR irradiation. In vivo fluorescence images showed that albumin nanoparticles accumulated in tumor tissues at a maximum amount at 24 h after intravenous administration. The generated hyperthermia upon NIR irradiation triggered drug release and caused tumor death [200].

## 6. Dual-Stimuli Responsive Nanomaterials Targeting Integrins

There is evidence suggesting that endogenous stimuli, such as pH, redox, and enzymes, and/or exogenous stimuli, such as temperature and magnet, are not sufficient for bench-to-bed translation of targeted stimuli-responsive nanomaterials [201]. To overcome this barrier, great effort has been made to prepare dual-stimulus responsive nanoparticles to offer unprecedented control over drug delivery and release, leading to superior antitumor potency in vitro and/or in vivo [202]. In this section, we discuss dual-stimuli responsive theranostic nanomaterials in the presence of integrin ligands.

### 6.1. pH-Redox (ROS)-Responsive Theranostic Nanomaterials

Among stimuli-responsive systems, pH and ROS stimuli are the most appreciated in cancer therapy. Their combination allows promising improvements in the stability of nanoparticles in vivo, as well as activation of drug release and enhanced tumor cell uptake [203]. The concentration of ROS in cancer cells is higher than that in normal cells [204]. The acidic pH present in the endosomal/lysosomal compartments (pH 4–5) is important for the development of pH-responsive nanocarriers. However, complete drug release may also be limited by H^+^ depletion in the absence of adequate supplementation [205]. Their combination leads to increase in vivo stability of nanoparticles, activation of drug release, and enhanced tumor cell uptake [203]. An example of this synergistic effect was reported by Bahadur et al. in 2012. They prepared polymeric nanoparticles with a disulfide-S-S-linkage, which presented suitable stability in the extracellular matrix. The researchers prepared cRGD-γ-poly(2-(pyridin-2-yldisulfanyl)ethyl acrylate)-γ-polyethylene glycol polymer and functionalized it with cRGD through a thiol-disulfide exchange reaction. Doxorubicin was encapsulated in the pore of the nanoparticles, and HCT-116 colon cancer cells were used for the experiments. The researchers observed enhanced cellular uptake and nuclear localization due to the presence of RGD and a fast release of the drug due to the acidic pH and redox potential of the cancer cells [206]. Another interesting approach was developed by Yao and co-workers who prepared a pH-ROS nanostimuli-responsive system. The nanocarrier was formed by 4-hydroxymethylphenylboronic acid pinacol ester as an ROS-responsive moiety, as well as cyclodextrin. The system was loaded with docetaxel. To enhance cellular uptake, nanoparticles were modified with cRGD. To fully release docetaxel (DTX), nanoparticles must be disassembled. After internalization into cancer cells by cRGD, the aldehyde is released, ROS oxidize the boron atom, inducing the breakdown of the borate ester, and then the chemical bond between cyclodextrin and the phenyl group is broken, resulting in the full release of the drug. The drug delivery system was tested in 4T1 breast cancer cells and showed substantial antitumor activity after 24 h. Activity was maintained after 48 h of incubation. In vivo, DTX/RGD@NP showed lower tumor volume and weight after 20 days of treatment demonstrating significant tumor-inhibition efficiency. The ability of DTX/RGD@NPs to inhibit lung metastasis of breast cancer was also investigated. Treatment with saline and the blank control resulted in a large number of metastatic nodules. While, after treatment with DTX/RGD@NPs, no metastatic clots were observed in the lung tissue of mice [20].

### 6.2. Temperature-pH-Responsive Theranostic Nanomaterials

An interesting dual approach is to use high temperatures and acidic conditions, both of which are present in the extracellular environment of the cancer cells [207]. For example, Kim and co-workers in 2023 prepared a thermosensitive polymer, N-isopropyl acrylamide-co-N-vinyl-imidazole-co-acrylic acid, based on mesoporous nanomaterial (MSN) coated with RGD sequence, pNIBIm-AA-RGD/MSN, and Dox embedded inside the pores of the MSN. The system consists of a pNIBIm-AA-RGD copolymer shell and an MSN as drug cargo with a diameter of 90 nm [208]. It has been demonstrated that the pNIBIm polymer is able to retain more drug at temperatures below the low critical solution temperature (LCST), whereas the drug is released from the polymer at higher temperatures. The acidic conditions in the cancer cell play a synergic effect with the high temperature for the drug-release [209]. The results of this study showed a rapid release of Dox equal to 90% at 40 °C and pH = 6.5 in 1 day, while at 40 °C and pH = 7.4 the release was equal to 85% in 2 days [208].

### 6.3. Light-Redox-Responsive Theranostic Nanomaterials

The dual effects of the light and the capacity to reduce glutathione (GSH) have been studied by several researchers. Wang and co-workers developed a gene delivery system using RGD-modified disulfide cross-linked short PEIs (DSPEI) functionalized on gold nanoparticles [210]. In this study, near-infrared (NIR) light was used, with an absorbance range of 650–900 nm. This provides deep tissue penetration and high spatial accuracy without damaging healthy tissues. They exploited the optical and electronic properties of gold nanoparticles (GNPs), where the plasmons present on the surfaces of the nanoparticles decay after excitation, leading to the transfer of excited electrons to the adsorbate before thermalization. Researchers covalently bonded to a carrier on the surfaces of the nanoparticles through an Au-S bond. Finally, a therapeutic molecule was loaded via a weaker interaction to allow the drug to be released after irradiation with a continuous-wave laser. Nanoparticles absorb energy, resulting in a reduction in the interaction between the nanocarrier and therapeutic molecules. This, in turn, leads to drug release at a specific site due to the presence of the RGD motif [210]. In another study, the effect of NIR exogenous stimuli combined with the pH-endogenous was studied by Zhou et al. in 2019 [211]. They prepared a system composed of an amphiphilic polymer PEG-Nbz-PAE-Nbz-PEG called HTMP, containing PEG (as a hydrophilic segment) and pH-sensitive poly(β-aminoester)s-PAE (as a hydrophobic segment) covalently linked via an o-nitrobenzyl (Nbz) linker. Another amphiphilic polymer, iRGD-PAE-iRGD (iPHT) was also prepared. These two parts were used to prepare a hybrid micelle called iPHM. Core-shell structured NaYF4:Yb/Tm@NaYF4 and up-conversion nanoparticles (UCNPs) were introduced to convert NIR to UV-vis, and doxorubicin was loaded into the nanoparticles to form iPHM@UCNP named iPUDN. According to the obtained results, iPUDN showed a different release of doxorubicin at several pH values. At pH 7.4, the system showed a slow release of Dox of approximately 27% in 48 h. Once the pH reached 5.5, Dox release increased to 66% at the same time. Following NIR irradiation, 80 amounts of Dox was released at pH 5.5, confirming the key role of NIR and pH in enhancing the Dox release from iPUDN. To evaluate the ability of iPUDN to penetrate, the authors worked on MCF-7 cells and demonstrated a remarkable uptake due to the presence of iRGD. The effect of NIR is to induce dePEGylation of iPUDN to facilitate exposure of iRGD to cells [211].

### 6.4. pH-Light-Responsive Theranostic Nanomaterials

The combination of pH and light could be an interesting approach to prepare dual-stimuli-responsive theranostics. For instance, Kuang et al. [212] prepared nanoparticles formed by gadolinium (Gd) and hafnium dioxide (HfO_2_), to confer high relativity, radio-enhancement ability, and biosafety [213,214]. The authors prepared a system called Gd2Hf2O7@PDA@PEG-Pt-RGD, where polydopamine (PDA) is a photothermal agent and RGD is an integrin ligand, while PEG is used for the biostability and cisplatin is used as a chemotherapeutic agent. Tests were conducted on A549R cells (adenocarcinomic human alveolar basal epithelial cells). The authors did not observe any cytotoxicity of Gd2Hf2O7@PDA@PEG-Pt-RGD after 24 h of incubation at different concentrations. However, after NIR irradiation, the cell viability decreased when the concentration of the nanoparticle system increased. The effects of hyperthermia include the denaturation of cytoplasmic proteins, inhibition of DNA repair, and disturbances in signal transduction. An enhanced release of cisplatin was observed when the pH was reduced from 7.4 to 6.5 ranging from 24.4 ± 0.3% to 46.1 ± 1.6% [212]. Another interesting nanoparticle system was prepared by He et al. in 2021 [215]. They prepared melanin-coated magnetic nanoparticles (MMNs) conjugated with an RGD sequence and co-loaded with Dox and ICG through π−π stacking interactions (RMDI). It was observed that the release of Dox was 26.5% at pH 6 whereas the drug release was 17.3% after 4 h of treatment at pH 7. After NIR irradiation for 5 min, the Dox release was enhanced. Accumulation of RDMI was observed after the treatment of glioblastoma cells (U87MG) owing to the presence of the RGD sequence on the outer shell of the nanoparticles. This cytotoxic effect resulted in a 40% reduction in cellular viability. Therefore, under NIR laser irradiation, the cytotoxic effect was higher owing to the synergistic effect of the photothermal-enhanced chemotherapy. In vivo study conducted on U87MG showed an enhanced temperature in the local tumor after laser irradiation followed by a complete photothermal destruction of tumor cells [215].

### 6.5. pH-Magnet-Responsive Theranostic Nanomaterials

The preparation of pH-magnet dual-responsive theranostics is another approach that has attracted considerable attention in recent years. Gao et al. developed polymeric micelles with the simultaneous ability of targeted drug delivery and imaging. They used DOX as a model anticancer drug that can be released from the micelles by a pH-responsive mechanism, cRGD as an αvβ3 integrin ligand, superparamagnetic iron oxide as a material for MRI detection, and PEG-block-poly(D,L-lactide) co-polymer as a model carrier polymer. cRGD-SPIO-DOX micelles with the size around 45–47 nm were synthesized, and in vitro tests were conducted on SLK endothelial cells over expressing αvβ3 integrin. These results demonstrated that the micelles have a promising therapeutic and diagnostic abilities [216]. In another study, Wu et al. designed a theranostic nanoparticle model. For this purpose, they utilized DOX as a model chemotherapeutic agent, RGD2 as a ligand for αvβ3 integrin, exceedingly small magnetic iron oxide nanoparticles (ES-MION) as a MRI contrast agents, and poly(ethylene glycol) methyl ether (mPEG) as a model polymer. The size of the DOX@ES-MION@RGD_2_@mPEG nanoparticles was approximately 13 nm, which can prolong the nanoparticle circulation. For in vitro testing, U-87 MG cells and MCF-7 cells were integrin αvβ3 positive and negative, respectively. The results demonstrated that at neutral pH, RGD2 can be hidden by mPEG, but can be exposed to tumor cells mildly acidic pH, which might improve cell targeting. Additionally, DOX is released at pH of approximately 5.5, which reduces the side effects of the drug in normal cells. In vivo tests performed on nude mice confirmed that the prepared nanoparticles could transfer the drug with high efficiency [217]. In another investigation, Subramanian et al. developed amphipathic chitosan-based targeted nanomicellar theranostic doxorubicin-superparamagnetic iron oxide nanoparticles (SPIONs). In this study, RGD was used as a ligand for targeting integrin and amphipathic chitosan for pH responsiveness. They used DOX and SPIONs as the model anticancer drugs and diagnostic components, respectively. The size was approximately 30–45 nm. In vitro studies were performed on Dox-resistant triple-negative breast cancer human cells (MDA-MB 231) and murine cells (4T1). The results indicated that Ab-CS-Dox-SPION micelles had maximum anti-migration activity with minimal scratch closure in both the cell lines. In vivo studies were carried out on female BALB/c mice where the intratumoral injection of nanomicelles showed higher anti-tumor effect and enhancement in MRI-T2 contrast simultaneously [218].

## 7. Stimuli-Responsive Nanomaterials Targeting Integrin for Tumor Diagnosis

The development of an early and efficient diagnostic tool has attracted great interest for the fight against cancer. To date, numerous technologies have been developed, including magnetic resonance imaging (MRI), positron emission tomography (PET), and single-photon emission computerized tomography (SPECT). However, they lack efficacy and selectivity in specific tissues. In 2018, a near-infrared (NIR) fluorescence imaging reduced glutathione (GSH)-activated probe (CyA-cRGD) was developed for a rapid and accurate cancer diagnosis, taking advantage of its high affinity for αvβ3 integrin (Figure 4) [219]. NIR is a non-invasive, inexpensive, and easy-to-perform system owing to its molecular fluorescence as an exogenous contrast agent. The probe, prepared by Yuan et al., showed a rapid and significant enhancement of the NIR fluorescence characteristics. Once the system was used in vivo, they observed a high efficiency for early-stage tumor diagnosis within 72 h of its implantation [219]. In 2020, Zhang et al. developed a contrast agent to be use in magnetic resonance imaging (MR) for hepatocellular carcinoma (HCC) [220]. The gadolinium-based (Gd) contrast agent was RWY-dL-(Gd-DOTA)_4_, the RWY motif was able to recognize the integrin α6. In addition, a PEG4 spacer was used to avoid steric hindrance of Gd-DOTA monoamide. Moreover, a lysine dendrimer was employed to increase the molar ratio of Gd-DOTA monoamide to the peptide. The authors used HCC-LM3 subcutaneous liver tumors and the analysis revealed that RWY-dL-(Gd-DOTA)_4_ can provide a signal three times higher than Ctrl-dL-(Gd-DOTA)_4_ [220]. Among external stimuli, the use of ultrasound has attracted great interest in recent years for the sensitive detection of intravascular targets [221]. Contrast ultrasound imaging is inexpensive, and permits real-time anatomical and molecular imaging. This technique has important applications in the imaging of tumor angiogenesis due to the presence of micron-scale microbubbles (MB) acting as contrast agents [222]. The most relevant MB contrast agent requires a hybrid low-high-mechanical-index (MI) pulse. A high MI was used to destroy the circulating MB after injection. In this context, Anderson et al. developed a cRGD-MB capable of working with low-MI pulses for a sensitive clinical imaging system. The cRGD-MB was tested on mouse model of mammary carcinoma and the researchers observed significantly enhanced contrast signals with a high tumor-to-background ratio [223].

## 8. Future Perspective and Challenges

The clinical translation of integrin stimuli-responsive theranostics depends on several crucial questions that must be answered. These questions can be categorized in two different levels. The first level is the factors associated with the integrin ligand. There are various integrin receptors and targeting ligands. In order to bench-to-bedside translation of integrin-decorated nanoplatforms, the integrin targeted sequence must be designed precisely to enhance the interaction of the ligand with desired receptors. The second point is the density and degree of conjugation which has considerable impact not only on the physicochemical properties of the theranostics agent but also on its pharmacokinetic characteristics such as blood circulation time, metabolism, and excretion. The conjugation degree and number of integrin ligands on the nanoplatform is a determining factor for the affinity of the delivery system to its receptor. Finding the optimum degree of conjugation is a crucial point for the future development of each nanoplatform. Since there is no universal formula to determine the optimum number of ligands on a nanoplatform, this number must be found case-by-case. Another major point is the availability of the ligand for integrin receptor following its conjugation on the platform structure. Successful conjugation of integrin ligands on a delivery system with optimum number does not necessarily lead to efficient targeted delivery. The 3D structure of the receptor and the conformation of the ligand following conjugation determine whether an efficient interaction between the receptor and its ligand occurs or not. In this case, the presence of appropriate linkers facilitates the successful interaction between the integrin receptor and its ligand. The length of the linker, its chemical composition, and the presence of different functional groups must be optimized too.

The second level of crucial question for the clinical translation of integrin-stimuli responsive theranostics is their properties following in vivo administration including pharmacokinetic characteristics and safety as well as their efficiency to accumulate in the desired cells, tissues, or organs. The conjugation of ligand on a nanoplatform not only changes the physicochemical properties of the ligand itself but also the nanoplatform too. These changes may have considerable impact on the pharmacokinetic properties of theranostics following the administration. Since the main route of administration is the injection such as intravenous administration, the interaction of nano-theranostics with blood component is another major concern for their clinical translation. It has been shown that the formation of protein corona around the nano-delivery systems changes their properties and inhibits their recognition by the receptors. The composition of soft and hard corona layers around the nano-theranostics depends on several personalized factors including age, gender, ethnicity, and other diseases in the patient. These factors change the composition of the protein corona leading to different pharmacological responses in different individuals. These inter-individual differences may hamper successful clinical translation of targeted nano-theranostics. Another major point for bench-to-bed translation of these systems is their safety following administration. Although the safety of RGD peptide has been shown in several studies, the toxicity concern related to each new sequence must be determined. In other words, the safety results of previously studied similar sequences cannot be extrapolated to the novel ones.

The last but not the least challenge for their clinical translation is that integrins are ubiquitous receptors, and can be found on normal cells too. This point raises the question on their in vivo specificity following systemic administration of integrin-targeted constructs. Therefore, an appropriate target-to-background ratio must be carefully considered.

## 9. Conclusions

Considering the fact that various number nano-based platforms are in clinical trials either for diagnostic, therapeutic, or theranostic purposes, the modification of such nanoplatforms is a prerequisite step for their bench-to-bedside translation. These modifications must include, but are not limited to, the improvement of carrier pharmacokinetic behaviors, such as prolonged circulation time and reduced premature release of the cargo as well as their biocompatibility and biodegradability. In this regard, transferring the cargo into a precise site of action, such as tumor cells, is one of the major obstacles hampering the wide clinical application of these platforms. The decoration of nanoplatforms with integrin ligands has attracted considerable attention. However, integrin decoration has certain limitations. Integrins are expressed in cancer and normal cells. Therefore, off-target effects are a major limitation of the integrin-decorated nanoplatforms. The integrin family consists of several receptors that have affinity for various targeting ligands. Thus, finding a specific ligand with the ability to target a particular subfamily is a necessary step towards the clinical translation of such theranostics. On the other hand, various approaches have been employed to improve pharmacokinetic characteristics of the platforms including PEGylation. Although this approach has shown great ability to increase the half-life and circulation time of the delivery system as well as the reduction in its opsonization by the immune system, there are several concerns regarding its combination with targeting strategies. PEG molecules may hide targeting ligands decorated on the surface of nanoplatforms and reduce their recognition by the specific receptors. Thus, the optimization of shielding agents such as PEG in terms of molecular weight and substitution degree is another point that must be considered for clinical translation.

The authors believe that the successful translation of integrin stimuli-responsive theranostics depends on the shoulder-to-shoulder development of two key factors: nano-theranostic platforms must be developed to ensure improved pharmacokinetic properties as well as promising toxicity, biocompatibility, and biodegradability. The second step is to concentrate on the integrin ligands. In this regard, it is suggested that integrin profiling be performed to find specific integrins overexpressed on particular target cancer cells. More precise ligand–receptor interactions reduce the off-target effects, increasing their potential for clinical applications. The combination of these strategies may create novel modalities for targeted stimuli-responsive diagnostic, therapeutic, and theranostic platforms as a prerequisite step for personalized medicine.

## Figures and Tables

**Figure 1 pharmaceutics-16-01441-f001:**
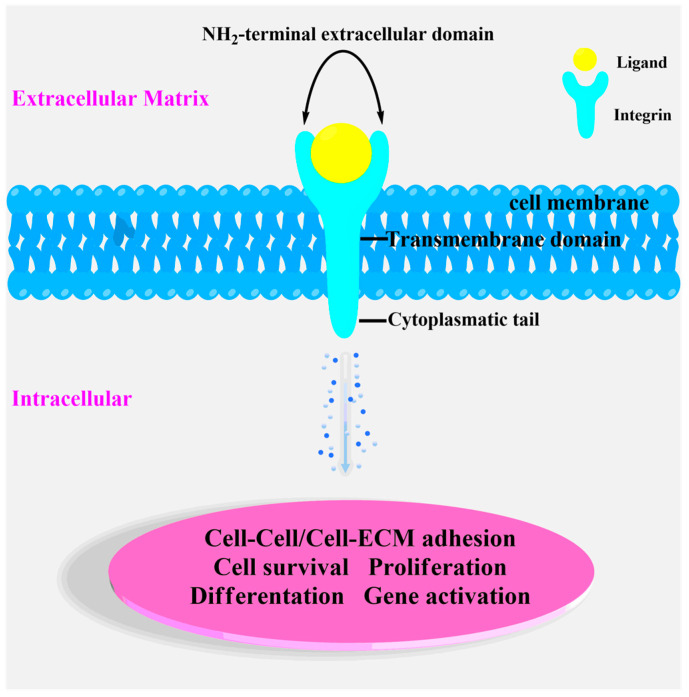
Schematic representation of integrins structure.

**Figure 2 pharmaceutics-16-01441-f002:**
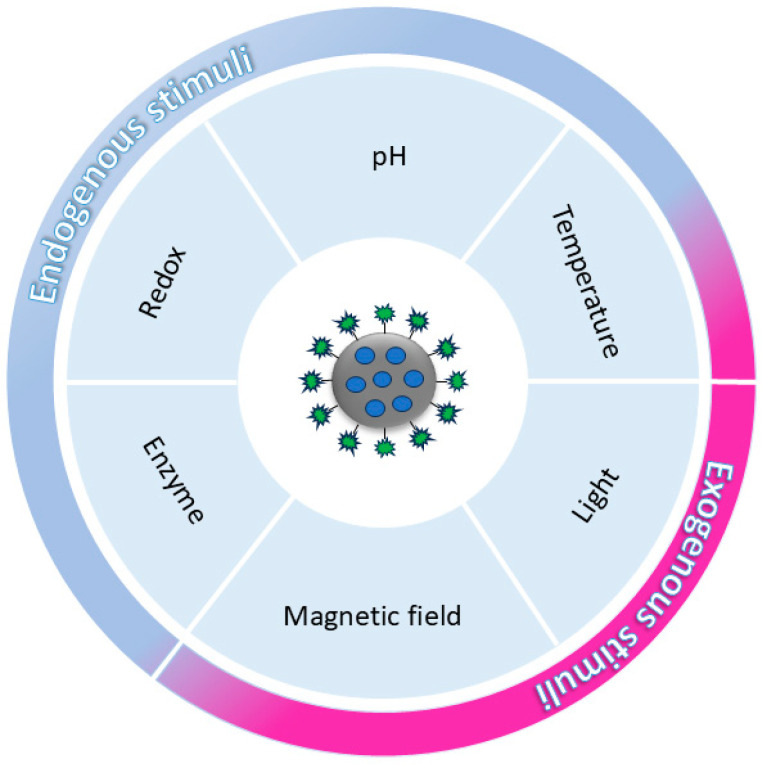
General representation of internal (endogenous) and external (exogenous) stimuli used to induce the release of cargo from the nanoparticles.

**Figure 3 pharmaceutics-16-01441-f003:**
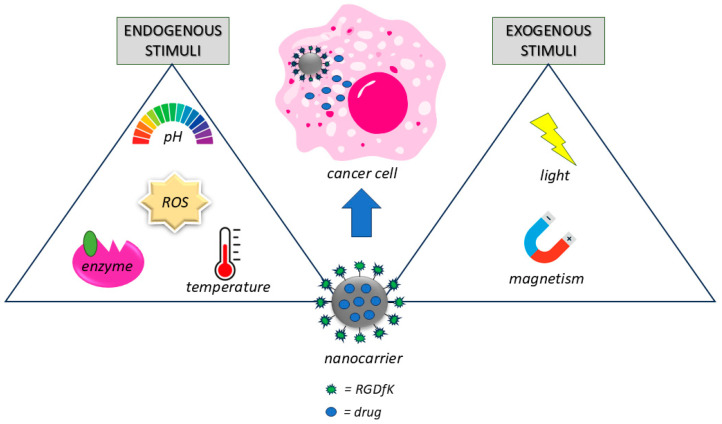
Schematic representation of the nanocarrier loaded with integrin ligand and encapsulated with drug. Stimuli-responsive nanocarrier triggers drug release from the nanoparticles into the cancer cell. On the left, the endogenous stimuli; on the right, the exogenous stimuli.

**Figure 4 pharmaceutics-16-01441-f004:**
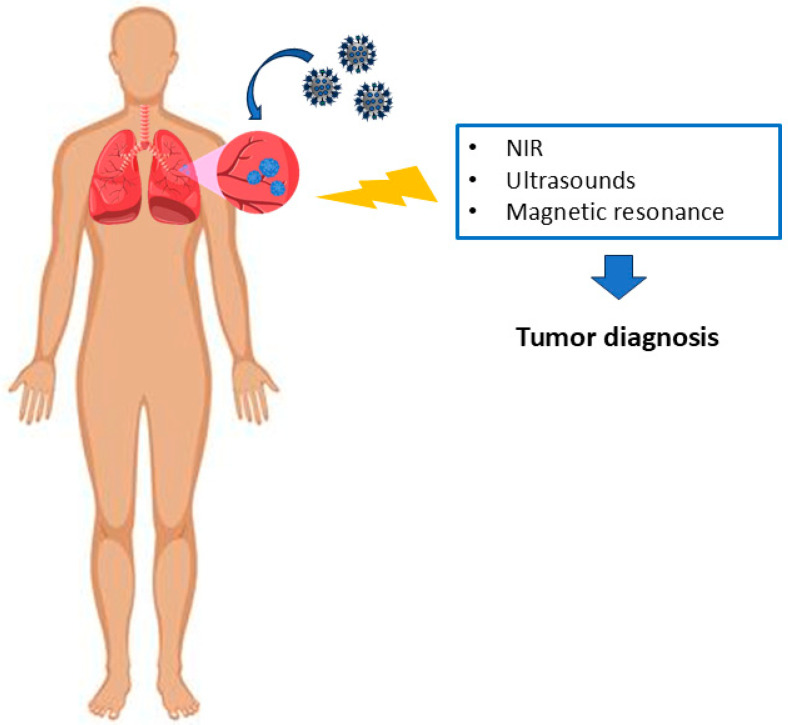
Schematic representation of stimuli-responsive nanomaterials targeting integrin for tumor diagnosis.

**Table 1 pharmaceutics-16-01441-t001:** Classification of Human G-protein-coupled receptors (GPCRs) on the basis of the phylogenetic studies and their structural and functional properties.

Class	Structural Properties	References
*Family A or rhodopsin-like receptors*	(a) An elliptic and cylindric shape; this is due to the arrangement of the 7-TMD which varies in length from 20 to 30 amino acid residues. (b) The dimensions of rhodopsin are approximately 35 × 48 × 75 Å with the longitudinal axis perpendicular to the membrane. (c) The presence of the dipeptide Gly-Pro induces a conformational change with an irregular shape of the TMD of the rhodopsin receptor.	[25,26,27,28]
*Family B1 or secretin receptor (SFR)*	(a) The secretin receptor family (SFR) is formed by 15 members which share between 21 and 67% sequence identity. (b) SFR contains, in the first and second extracellular loop of the TMD regions, conserved cysteine residues. (c) The ligand interacts with secretin receptor by three binding domains, the proximal and the juxta membrane region of the N-terminus, the extracellular loops, together with TM6.	[29,30]
*Family B2 or Adhesion receptors*	(a) The adhesion class consists of an extracellular domain, the GPCR autoproteolysis-inducing (GAIN) domain, a 7TM domain, and an intracellular domain. (b) They are situated on the cell surface as non-covalent heterodimer and consist of two subunits α and β generated by autoproteolytic event. (c) The presence of an extracellular domain much larger than other GPCRs facilitates the cell-cell interaction and cell-matrix interaction.	[31]
*Family C or metabotropic glutamate receptors* (GRM)	(a) The presence of a large extracellular domain containing the orthosteric sites which forms constitutive dimers with unique activation systems. (b) The presence of an unusually large extracellular domain, an intracellular carboxyl terminal (C-terminal) domain, and a heptahelical TMD. (c) The extracellular domain of GRM is rich in cysteine and Venus flytrap module (VFT) and the presence of only allosteric site in the TM domain of GRM; the domains of GRM provide many ligand sites of action.	[32]
*Family F or frizzled (FZD) and smoothened (SMO) receptors*	(a) Smoothened (SMO) receptors present a cysteine-rich domain (CRD) in the extracellular part and are involved in embryonic development, body shaping, and maintenance and regeneration of adult stem cells. (b) They are fundamental to mediate Hedgehog (Hh) signaling pathway, and its activity is blocked when it binds to the protein 12TM Patched1 (PTCH). (c) FZDs are the receptors of Wnt proteins and mediate the Wnt/β–catenin signaling pathway. FZD is activated by Wnt which also triggers the co-receptor low-density-lipoprotein-related protein (LRP) 5/6 to recruit intracellular partners to form the central signalosome. This signaling leads to stabilization of the β–catenin and translocation into the nucleus for upregulating target genes. FDZs are oncoproteins and they are involved in cancer and other pathologies.	[33,34,35,36]

**Table 2 pharmaceutics-16-01441-t002:** Examples of stimuli-responsive nanocarriers reported in clinical trials.

Stimulus	Nanocarrier	Therapeutic Agent	Cancer	Clinical Phase	Reference
Enzyme	Liposomes	Cisplatin	Breast cancer, Advanced or refractory solid tumor, Metastatic Prostate cancer, Skin cancer	I, II	NCT01861496
Magnetic	Iron and carbon	Doxorubicin	Hepatocellular carcinoma	I, II	NCT00034333
Magnetic	Iron and carbon	Doxorubicin	Liver metastasis	I, II	NCT00041808
Magnetic	Iron oxide magnetite	Iron oxide nanoparticles	Prostate cancer	I	NCT02033447
Magnetic	Iron and carbon	Doxorubicin	Unresectable hepatocellular carcinoma	I, II	NCT02033447
pH	Polymeric micelles	Epirubicin	Sarcoma, Metastatic sarcoma, Solid tumor, Soft tissue sarcoma	I, II	NCT03168061
Temperature	Liposomes	Doxorubicin	Recurrent regional breast cancer	I, II	NCT00826085
Temperature	Liposomes	Doxorubicin	Hepatocellular carcinoma	III	NCT02112656
Temperature	Liposomes	Doxorubicin	Pediatric refractory solid tumor	I	NCT02536183
Temperature	Liposomes	Doxorubicin	Liver tumor	I	NCT02181075
Temperature	Liposomes	Doxorubicin	Adenocarcinoma, Breast carcinoma, Non-small-cell lung cancer, Painful bone metastases, Small-cell lung cancer	II	NCT01640847

**Table 3 pharmaceutics-16-01441-t003:** Examples of stimuli-responsive nanocarriers and their properties.

Stimuli	Nanomaterial	Component	Therapeutic Agent	Cancer Type	Modality	Ref.
pH	Nanoparticles	Lactobionic acid	Sorafenib, Curcumin	Hepatocellular Carcinoma	Therapeutic	[142]
pH	Liposome	SS-cleavable and pH-activated lipid-like material and vitamin E	-	-	Therapeutic	[143]
pH, Redox	Mesoporous silica nanoparticles	Chitosan-folate	Doxorubicin	Breast cancer	Therapeutic	[144]
pH	Polymeric nanoparticles	PLGA ^a^	Doxorubicin	-	Therapeutic	[145]
pH	Nanodots	PEG_5k_-PAE_10k_	Quercetin	Liver cancer	Theranostic	[146]
pH	Nanoparticles	Acetalated dextran	Platinum	-	Theranostic	[147]
pH	Nanoparticles	Zinc oxide	Quercetin	Breast cancer	Therapeutic	[148]
pH	Lipid-polymer hybrid nanoparticles	PEG ^b^	Carboplatin, Paclitaxel	Cervical cancer	Therapeutic	[149]
Enzyme	Polymeric nanoparticle	Saccharide	Paclitaxel	-	Theranostic and imaging	[150]
Enzyme	Mesoporous silica nanoparticles	Hyaluronic acid	5-fluorouracil	Colon cancer	Therapeutic	[151]
Enzyme, pH	Polymeric nanoparticles	PLGA, PEI ^c^, dimethyl maleic anhydride	Docetaxel	Breast cancer	Therapeutic	[152]
Enzyme	Nanoparticles	Poly(ethylene glycol)-*b*-poly(L-tyrosine)	Doxorubicin	Colorectal cancer	Therapeutic	[153]
Enzyme	Nanoparticles	Pep-Pt-P	Oxaliplatin,	-	Therapeutic	[154]
Enzyme	Nanocomposites	Guar gum	5-fluorouracil	Colorectal cancer	Therapeutic	[155]
Enzyme	Nanoparticles	mPEG-Peptide-PCL	Curcumin	Lung tumors	Therapeutic	[156]
Magnetic	Nanoparticles	mitochondria-targeting peptide and Fe_3_O_4_	-	-	Therapeutic	[157]
Enzyme	Metal-organic frameworks (MOF)	-	Doxorubicin	-	Therapeutic	[158]
Temperature	Liposomes	DPPC ^d^, MPPC ^e^	Tamoxifen, Imatinib	Breast cancer	Therapeutic	[159]
Temperature	Manganese oxide doped carbon dot	Fe^3+^ and biothiols	-	Liver cancer	Imaging	[160]
Temperature, pH, Magnetic	Nanocomposite particles	Fe_3_O_4_	Methotrexate	Breast cancer	Therapeutic	[161]
Temperature	Nanoparticles	PEGylated polyaspartamide derivative	Paclitaxel	-	Therapeutic	[162]
pH, Temperature	Magnetic nanoparticles	Folic acid	Doxorubicin	-	Therapeutic	[163]
Magnetic, Light	Nanoagent	mPEG-PLGA-PLL	Iron oxide, cytosine-phosphate-guanine oligodeoxynucleotide	Breast cancer	Theranostic	[164]
Magnetic, Temperature	Mesoporous Silica Nanoparticles	Iron oxide, PEG, Isopropyl acrylamide, Hydroxymethyl acrylamide, Methylenebis acrylamide	Doxorubicin	Lymphoma	Therapeutic	[165]
Magnetic	Nanocomposite	Iron ions	Cisplatin and Methotrexate	Colon cancer	Therapeutic	[166]
Magnetic, ultrasound	Nanobubble system	DPPC, DOPC ^f^, cholesterol, Fe^2+^, Fe^3+^	Pemetrexed, Pazopanib	Non-small-cell lung cancer	Therapeutic	[167]
Magnetic	Nanocomposite	Fe_3_O_4_	Doxorubicin	-	Therapeutic	[168]
Magnetic	core/shell nanoparticle	Iron oxide nuclei	Gemcitabine	Breast Cancer	Therapeutic	[169]
Magnetic	Nanoparticles	Iron oxide, Calcium phosphate, PEG,	SiRNA	Breast cancer	Therapeutic	[170]
Light	Hydrogel	Green cyanine dye	Doxorubicin	Oral squamous cell carcinoma	Therapeutic	[171]
Light	Microneedle and gold nanorod	Hyaluronic acid	Indocyanine green	Skin cancer	Therapeutic	[172]
Light	Nanoparticles	Bovine serum albumin	BODIPY	4T1 cancer cells	Theranostic	[173]
Light	Hydrogel	Hemoglobin, PEG	-	A549 lung cancer cells	Therapeutic	[174]
Light, Oxidation	Nanohybrids	Polyamidoamine-Poloxamer 188	Indocyanine green, Graphene oxide	-	Therapeutic	[175]
Light	Nanoparticles	Ferrocenecarboxylic acid	Cisplatin, Indocyanine green	-	Therapeutic	[176]
Light	Mesoporous nanoparticle	Polydopamine, 1-tetradecanol	Epigallocatechin-3-gallate, Diallyl trisulfide, Indocyanine green	-	Therapeutic	[177]
Light	Polymeric micelles	Poly(ethylene glycol)-block-poly-l-lysine	Docetaxel	-	Therapeutic	[178]

^a^ D,L-lactic-co-glycolic acid, ^b^ Poly ethylene glycol, ^c^ polyethyleneimine, ^d^ 1, 2-dipalmitoyl-sn-glycero-3-phosphocholine, ^e^ Monopalmitoyl-2-hydroxy-snglycero-3-phosphocholine, ^f^ 1,2-dioleyl-sn-glycero-3-phosphatidyl choline.

**Table 4 pharmaceutics-16-01441-t004:** Examples of stimuli-responsive nanocarriers targeting integrin receptors in cancer.

Stimulus	Targeted Integrin	Nanomaterial	Therapeutic Agent	Cancer Type	Ref.
Temperature	α6β4 and αvβ3	Liposome	Doxorubicin	Breast	[184]
Enzymatic	αvβ3	Cholesterol/DOPE/DSPC/DSPE-(PEO)4-cRGDfK/DSPE-mPEG20	Doxorubicin	Pancreatic, renal	[185]
Light	α4β1 and αvβ3	Polymer NP	Chlorine e6	Ovarian	[186]
Enzymatic	αvβ3 and αvβ5	Micelles	(1,2-diaminocylohexane)platinum(II)	Melanoma	[187]
Enzymatic	αvβ3 and αvβ6	Micelles	Paclitaxel	Glioma	[188]
pH	αvβ3 and αvβ5	Micelles	Epirubicin	Glioblastoma multiforme	[189]
Temperature	α1β1, α2β1 and α11β1	Dendrimers	Doxorubicin	Fibroblast	[190]
Enzymatic	α5b1 and αvb3	Poly(ethylene glycol) (PEG) grafted chitosan–poly(ethylene imine) hybrid NP	siRNA	Non-small-cell lung carcinoma	[191]
pH	αvb6	Polymersomes	Paclitaxel	Colon	[192]
Enzymatic	αvb3	Perfluorocarbon-NP	Fumagillin	Adenocarcinoma	[193]

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
