# Peer review of "Integrin-Specific Stimuli-Responsive Nanomaterials for Cancer Theranostics"

_pharmaceutics, 2024, doi:10.3390/pharmaceutics16111441_

Round 1
Reviewer 1 Report
Comments and Suggestions for Authors
The article is of good quality and clear. I recommend this paper to be published in the journal. Here are some minor suggestions:
1. The authors emphasize the advantages of integrin-specific stimuli-responsive nanomaterials in tumor treatment, but the potential side effects, toxicity, or challenges should be also discussed.
2. In Page 5, the article reports examples of stimuli-responsive nanocarriers reported in clinical trials. Could you provide a more in-depth outlook on its prospects in clinical applications?
3. In Sections 4-5, the authors describe current research. It would be beneficial to add the shortcomings of current studies and areas for improvement.
4. There is a lack of recent literature citations. Authors also should cite some literatures published between 2023 and 2024. “Conventional chemo and radiotherapy lead to significant side effects due to lack of selectivity, while surgery might be invasive for patients. Hence, remarkable efforts are underway to increase the efficiency while maintaining the side effects as low as possible. (DOI: 10.1002/adhm.202303626)”; “The main challenge for researchers is to develop stimuli-responsive nanomaterials capable of exploiting internal and/or external stimuli such as pH, reactive oxygen species (ROS), redox, enzymes present in the tumor environment and external stimuli including light, proximity infrared (NIR) and magnetic field. (DOI: 10.1021/acsbiomaterials.4c00388; DOI: 10.1007/s40843-024-2855-2)”; “GSH levels in cytoplasm is about 1-10 mM, that is higher than in extracellular environment (2-20 µM). (DOI: 10.1016/j.phrs.2024.107150)”; “Another strategy for the design of drug delivery systems for cancer therapy is adopting enzyme responsive nanocarriers. In human body, biological activity and metabolic processes are carried out by enzymes, which have high specificity for their substrates. (DOI: 10.1016/j.nantod.2024.102421)”.
5. “Significant angiogenesis, different metabolic pathways, and rapid proliferation as well as high oxidative stress, high expression of some enzymes and weak acidity make tumor microenvironment much different from the other parts of the .” should be corrected. It is recommended that the authors improve the quality of language.
6. The style of reference citation should be corrected according to the requirements of the journal.
Author Response
- The authors emphasize the advantages of integrin-specific stimuli-responsive nanomaterials in tumor treatment, but the potential side effects, toxicity, or challenges should be also discussed.
In order to address the reviewer’s comment, a section entitled “Future perspective and challenges” were added as follows:
Future perspective and challenges
The clinical translation of integrin stimuli-responsive theranostics depends on several crucial questions that must be answered. These questions can be categorized in two different levels. The first level is the factors associated with the integrin ligand. There are various integrin receptors and targeting ligands. In order to bench-to-bedside translation of integrin-decorated nano-platforms, the integrin targeted sequence must be designed precisely to enhance the interaction of the ligand with desired receptors. The second point is the density and degree of conjugation which has considerable impact not only on the physicochemical properties of the theranostics agent but also on its pharmacokinetic characteristics such as blood circulation time, metabolism and excretion. The conjugation degree and number of integrin ligands on the nanoplatform is a determining factor for the affinity of the delivery system to its receptor. Finding the optimum degree of conjugation is a crucial point for the future development of each nano-platform. Since there is a not a universal formula to determine the optimum number of ligands on a nanoplatform, this number must be found case-by-case. Another major point is the availability of the ligand for integrin receptor following its conjugation on the platform structure. Successful conjugation of integrin ligands on a delivery system with optimum number does not necessarily lead to efficient targeted delivery. The 3D structure of the receptor and the conformation of the ligand following conjugation determine whether an efficient interaction between the receptor and its ligand occurs or not. In this case, the presence of appropriate linkers facilitates the successful interaction between the integrin receptor and its ligand. The length of the linker, its chemical composition and the presence of different functional groups must be optimized too.
The second level of crucial question for the clinical translation of integrin-stimuli responsive theranostics is their properties following in vivo administration including pharmacokinetic characteristics and safety as well as their efficiency to accumulate in the desired cells, tissues or organs. The conjugation of ligand on a nanoplatform not only changes the physicochemical properties of the ligand itself but also the nano platform. These changes may have considerable impact on the pharmacokinetic properties of theranostics following the administration. Since the main route of administration is the injection such as intravenous administration, the interaction of nano-theranostics with blood component is another major concern for their clinical translation. It has been shown that the formation of protein corona around the nanodelivery systems changes their properties and inhibits their recognition by the receptors. The composition of soft and hard corona layers around the nanotheransotics depends on several personalized factors including age, gender, ethnicity and other diseases in the patient. These factors change the composition of the protein corona leading to different pharmacological responses in different individuals. These inter-individual differences may hamper successful clinical translation of targeted nano-theranostics. Another major point for bench-to-bed translation of these systems is their safety following administration. Although the safety of RGD peptide has been shown in several studies, the toxicity concern related to each new sequence must be determined. In other words, the safety results of previously-studied similar sequences cannot be extrapolated to the novel ones.
The last but not the least challenge for their clinical translation is that integrins are ubiquitous receptors, and can be found on normal cells too. This point raises the question on their in vivo specificity following systemic administration of integrin-targeted constructs. Therefore, an appropriate target-to-background ratio must be carefully considered.
- In Page 5, the article reports examples of stimuli-responsive nanocarriers reported in clinical trials. Could you provide a more in-depth outlook on its prospects in clinical applications?
Dear reviewer thanks for your comment. We have revised in more depth and the following paragraph was added:
Liposomes are considered promising and versatile drug vesicles and used to designed a smart liposomal system able to respond at internal or external stimuli. They are formed by natural phospholipids that mimic the properties of biological membranes. By incorporating a polyethylene glycol (PEG) coating they have revealed a long circulation, liposomal formulations are able to improve the pharmacokinetics and pharmacodynamics of the drug, reducing its toxicity. Finally, the small size allows to promote the permea-tion and retention (EPR) effect, allowing passage into blood vessels and accumulating in tumor tissues. An interesting approach in clinical trials is liposomal cisplatin formu-lation (LiPlaCis), a drug delivery system formed to liposomes nanocarriers loaded with cisplatin. The system is developed to be degraded by secretory phospholipase A2 (PLA2), an enzyme presents in high concentrations in different cancer type and used for triggered drug-released into the organs/tissues. LiPLaCis shows greater distribution and reduced side effects when compared to the free drug. Among the developed nanoparticles, magnetic nanoparticles are developed. In presence of an alternative magnetic field, they generate local hyperthermia to trigger drug release and tumor ablation. The interaction between magnetic nanocarriers and magnetic field facilitates magnetically driven accu-mulation of nanocarriers in tumors. Biological tissues are transparent to magnetic fields, allowing magnetic targeting to be followed remotely and non-evasively. With this system, magnetic targeting is more flexible and does not depend on specific receptor ex-pression. Iron oxide (MION) and maghemite (gFe2O3) nanoparticles exhibit low tox-icity, high biocompatibility and thermal effects under different applied magnetic fields and small size, are in clinical trials and show interesting results for the treatment of hepato-cellular carcinoma and prostate cancer. Another interesting approach, is to use poly-meric micelles. These nanoparticles show the advantages to present a robust core–shell structure, kinetic stability and ability to solubilize hydrophobic drugs. These nanoparti-cles are able to release the drug in the acidic environment of the endosomal or lysosomal compartments of the target tumor cell taking advantage the different pH present in the normal cells.
- In Sections 4-5, the authors describe current research. It would be beneficial to add the shortcomings of current studies and areas for improvement.
Dear Reviewer, thanks for your comment. We have introduced the following sentence:
The use of nanoparticles for cancer treatment is increasing, but nanomaterials have an in-herent toxicity that can trigger accumulation of metals, used for nanoparticle synthesis, within tissues, side effects and drug release at a wrong site. The need to develop in-novative strategies to reduce nanotoxicity is the challenge that must be addressed to use nanomaterials safely.
- There is a lack of recent literature citations. Authors also should cite some literatures published between 2023 and 2024. “Conventional chemo and radiotherapy lead to significant side effects due to lack of selectivity, while surgery might be invasive for patients. Hence, remarkable efforts are underway to increase the efficiency while maintaining the side effects as low as possible. (DOI: 10.1002/adhm.202303626)”; “The main challenge for researchers is to develop stimuli-responsive nanomaterials capable of exploiting internal and/or external stimuli such as pH, reactive oxygen species (ROS), redox, enzymes present in the tumor environment and external stimuli including light, proximity infrared (NIR) and magnetic field. (DOI: 10.1021/acsbiomaterials.4c00388; DOI: 10.1007/s40843-024-2855-2)”; “GSH levels in cytoplasm is about 1-10 mM, that is higher than in extracellular environment (2-20 µM). (DOI: 10.1016/j.phrs.2024.107150)”; “Another strategy for the design of drug delivery systems for cancer therapy is adopting enzyme responsive nanocarriers. In human body, biological activity and metabolic processes are carried out by enzymes, which have high specificity for their substrates. (DOI: 10.1016/j.nantod.2024.102421)”.
Dear Reviewer, thanks for your comment. We have introduced the literatures into the manuscript.
- “Significant angiogenesis, different metabolic pathways, and rapid proliferation as well as high oxidative stress, high expression of some enzymes and weak acidity make tumor microenvironment much different from the other parts of the .” should be corrected. It is recommended that the authors improve the quality of language.
Dear Reviewer, your comment is correct. We have revised and corrected the sentence.
- The style of reference citation should be corrected according to the requirements of the journal.
Dear Reviewer, we have corrected the reference in the manuscript as requested by the journal.

Reviewer 2 Report
Comments and Suggestions for Authors
This work by Tehri et al comprehensively reviewed the stimuli responsive nanomaterials for cancers. The work is interesting. However, I would recommend some improvements based on the following points that would enhance the impact of this work.
1. Introduction is limited. It should be elaborative.
2. Most sections do not include any critical discussion and the works are just summarized one after another. The authors should focus on this vital aspect of the review. They should briefly examine and argue based on the reported studies.
3. More figures could be inserted to understand the concept easily as needed.
4. The challenges and future perspectives are critical parts of any review. The authors need to discuss them in a separate section.
5. I am not sure why the references are shown in roman numbers. They should be changed to English numbers for convenience.
Comments on the Quality of English Language
The English is fine.
Author Response
This work by Tehri et al comprehensively reviewed the stimuli responsive nanomaterials for cancers. The work is interesting. However, I would recommend some improvements based on the following points that would enhance the impact of this work.
- Introduction is limited. It should be elaborative.
The reviewer is right. It order to address the comment, the Introduction section was revised and the following paragraphs were added:
“Stimuli-responsive nanomaterials have shown passive targeting characteristics in vascularized solid tumors due to the enhanced permeability and retention (EPR) effect. However, recent studies have shown that EPR is not the only way for non-targeted nanomaterials to accumulate in the tumor sites. The complexity of tumor microenvironment and patient-to-patient variations in terms of gender, age and other diseases have led researchers to seek novel alternative strategies in order to transfer the payloads into their precise site of action. One of these successfully-tested approaches is targeting the tumor microenvironment using ligand-mediated strategies.”
“Integrin ligand-conjugated nanomaterials are internalized via clathrin or caveolin-1 internalization pathways rapidly. Since efficient cancer treatment might be hampered due to different resistance mechanisms, the receptor-mediated endocytosis leads to the intracellular delivery of payloads resulting in efficient therapeutic effects. In addition, there have been several reports on the application of integrin ligands as targeting moieties to transfer various types of delivery systems to integrin-overexpressed cells. The effects of integrin receptors in tumor growth, metastasis and angiogenesis have attracted great attention not only to use them as a target for cancer therapy but transferring various payloads into the cells over-expressing these receptors. The integrin-targeted nanomaterials may have several advantages including i) the reduction of non-specific interactions with normal cells, tissues and organs which subsequently leads to lower side-effects. ii) the accumulation of payloads in the precise site of action, higher concentrations of the drug in the target cell or tissue and subsequently the reduction of administrated primary dose. iii) the reduction of chemo-resistance due to the opportunity to deliver other therapeutic agents with the capability to overcome multi-drug resistance.”
- Most sections do not include any critical discussion and the works are just summarized one after another. The authors should focus on this vital aspect of the review. They should briefly examine and argue based on the reported studies.
The reviewer is right. There are several challenges and limitations in the reported studies. Therefore, a new section entitled “Future perspective and challenges” was added at the end of the manuscript to discuss different aspect and limitations of these studies. This section summarizes main concerns regarding further development of integrin-specific stimuli-responsive theranostics towards the clinic.
The clinical translation of integrin stimuli-responsive theranostics depends on several crucial questions that must be answered. These questions can be categorized in two different levels. The first level is the factors associated with the integrin ligand itself such as conjugation degree, orientation of ligand on the structure of delivery system and the presence of linker. The second level of crucial question for the clinical translation of integrin-stimuli responsive theranostics is their properties following in vivo administration including pharmacokinetic characteristics and safety as well as their efficiency to accumulate in the desired cells, tissues or organs. All of these aspects have been discussed in the recently-added section entitled “Future perspective and challenges”
- More figures could be inserted to understand the concept easily as needed.
Dear Reviewer, we have added the figures 2,3 and 4.
- The challenges and future perspectives are critical parts of any review. The authors need to discuss them in a separate section.
The reviewer is right. In order to address the reviewer’s comment, a section entitled “Future perspective and challenges” were added to the manuscript. Please see point 1(First Reviewer).
- I am not sure why the references are shown in roman numbers. They should be changed to English numbers for convenience.
Dear Reviewer, we have corrected the reference in the manuscript as requested by the journal.

Reviewer 3 Report
Comments and Suggestions for Authors
Dear authors, I read with interest your manuscript titled "Integrin-specific stimuli-responsive nanomaterials for cancer theranostics". This is a good paper describing the progress that has been made concerning the integrin-specific stimuli-responsive nanomaterials for cancer theranostics. I recommend the manuscript to be published after minor revision. I have some suggestions/comments/questions that may contribute to the work:
-Reference style should be checked (e.g. In the text, reference numbers should be placed in square brackets [] and placed before the punctuation, for example [1], [1-3] or [1,3].)
- A reference must be added for the first sentence after the introduction section.
- "conventional chemo and radiotherapy lead to significant side effects due to lack of selectivity". What are side effects and why do they occur?
- 4.2.1. Light-responsive nanocarriers - more examples should be added.
- The advantages/limitations of the integrin-specific stimuli-responsive nanomaterials vs the other NPs should be highlighted.
Author Response
Dear authors, I read with interest your manuscript titled "Integrin-specific stimuli-responsive nanomaterials for cancer theranostics". This is a good paper describing the progress that has been made concerning the integrin-specific stimuli-responsive nanomaterials for cancer theranostics. I recommend the manuscript to be published after minor revision. I have some suggestions/comments/questions that may contribute to the work:
-Reference style should be checked (e.g. In the text, reference numbers should be placed in square brackets [] and placed before the punctuation, for example [1], [1-3] or [1,3].)
Dear Reviewer, we have corrected the reference in the manuscript as requested by the journal.
- A reference must be added for the first sentence after the introduction section.
Dear Reviewer, we have corrected the reference in the manuscript as requested by the journal.
- "conventional chemo and radiotherapy lead to significant side effects due to lack of selectivity". What are side effects and why do they occur?
Dear Reviewer, thanks for your comments. We have added the following paragraph:
The cardiovascular system, kidneys, liver and lungs are the organs in which chemotherapy drugs and radiation induced by radiotherapy accumulate the most. This accumulation leads to serious side effects such as cardiotoxicity, ephrotoxicity, hepatotoxicity and pulmonary fibrosis. Most drugs have DNA in proliferating cells as a biological target. Consequently, tissues such as bone marrow and gastrointestinal tract, being rich in highly proliferating cells, lead to the formation of side effects such as gastrointestinal toxicity, myelosuppression and immunosuppression. Chemotherapy drugs and radiation also cause oxidative stress and inflammatory response because they present metabolic instability and can induce the formation of secondary tumors.
- 4.2.1. Light-responsive nanocarriers - more examples should be added.
Dear Reviewer, thanks for your comments. We have added the following paragraph:
An interesting approach is proposed to Diaz and co-workers. They used the laser-induced thermal mechanism to give de-hybridization of DNA or RNA sequences from gold nanoparticles. The nanoparticles are loaded with complex hybridizing molecules: a sin-gle-stranded complementary DNA, a DNA duplicator, and an abasically modified DNA duplicator. DNA duplicators show a different melting point and when light hits the nanoparticles, causing an increase in temperature, it allows the release of the labeled DNA duplicator in the cancer cells. The system offers the possibility of greater control of drug release. Chen et al. proposed to use red-light-responsive metallopolymer nanocarriers named PolyRuCHL formed to: hydrophilic poly(ethylene glycol) (PEG) block and a hydrophobic ruthenium (Ru)-containing block as red-light responsive linked with chloram-bucil (CHL). PolyRuCHL is loaded with DOX. The advantage of red light is the ability to pen-etrate deeply into tissue for PTT. Red-light irradiation induced a cleavage of Ru-CHL triggering its and DOX release from the nanoparticles. The authors observed a synergic effect to inhibit the growth and multidrug resistance in breast cancer cell line MCF-7R. The approach used by Fan and co-workers draws inspiration from the properties of liquid metals (LMs) that exhibit excellent photothermal conversion efficiency, generating heat under NIR laser irradiation. To prepare LMs, they used a gallium–indium eutectic alloys (EgaIn) which shows excellent combination of thermal-conductivity, transformability and a high biocompatibility. The authors prepared a poly(NIPAm-co-MBA) hydrogel (PNM) contained with LM droplets and encapsulated it with DOX to form the final system PNM/LM/DOX. After NIR irradiation, the temperature of the system rises above the lower critical temperature of the solution, which causes the hydrogel to change shape and size. The hydrogel shrinks, inducing a simultaneous release of the aqueous solution and DOX. This controlled release can reduce the amount of drug into normal tissues and thus the side effects.
- The advantages/limitations of the integrin-specific stimuli-responsive nanomaterials vs the other NPs should be highlighted.
The reviewer is right. In order to address the reviewer’s comment, a section entitled “Future perspective and challenges” were added to the manuscript.

Reviewer 4 Report
Comments and Suggestions for Authors
This manuscript provided a review for integrin-specific-stimuli nanocarriers for cancer theranostics. Several types of stimuli sources such as enzyme, magnetic, pH, temperature, etc. had been addressed and discussed. The reported findings from the literature of other workers were then summarized in the Tables. The paper gave us a comprehensive understanding of various types of stimuli-responsive nanocarriers for cancer treatments. However, I found that an important topic was missing in this manuscript, i.e. the use of liquid gallium nanoparticles for theranostics. So the authors need to incorporate such a topic in their manuscript. From browsing the webs, I found several important published papers relating liquid gallium and alloy nanoparticles conjugated with biomolecules for cancer treatments. They were listed as follows,
(1) Kulkarni et al. in Naanomedine: Nanotechnology, Biology and Medicine, Vol. 26, June 2020, 102175. Liquid metal based theranostic nanoplatforms.......
(2) Fu et al. in Journal of Materials Science & Technology, 142(2023)22-33. Gallium-based liquid metal micro/nanoparticles for photothermal cancer therapy.
(3) Liu et al. in Advanced Health Materials, 2022, doi: 10.1002/adhm.202102584.
(4) and much more reported papers in the literature.
Author Response
This manuscript provided a review for integrin-specific-stimuli nanocarriers for cancer theranostics. Several types of stimuli sources such as enzyme, magnetic, pH, temperature, etc. had been addressed and discussed. The reported findings from the literature of other workers were then summarized in the Tables. The paper gave us a comprehensive understanding of various types of stimuli-responsive nanocarriers for cancer treatments. However, I found that an important topic was missing in this manuscript, i.e. the use of liquid gallium nanoparticles for theranostics. So the authors need to incorporate such a topic in their manuscript. From browsing the webs, I found several important published papers relating liquid gallium and alloy nanoparticles conjugated with biomolecules for cancer treatments. They were listed as follows,
(1) Kulkarni et al. in Naanomedine: Nanotechnology, Biology and Medicine, Vol. 26, June 2020, 102175. Liquid metal based theranostic nanoplatforms.......
(2) Fu et al. in Journal of Materials Science & Technology, 142(2023)22-33. Gallium-based liquid metal micro/nanoparticles for photothermal cancer therapy.
(3) Liu et al. in Advanced Health Materials, 2022, doi: 10.1002/adhm.202102584.
(4) and much more reported papers in the literature.
Dear Reviewer, thanks for your comments. We have introduced and commented the following literature:
Fan, L.L.; Sun, X.Y.; Wang, X.L.; Wang, H.Z.; Liu, J. NIR laser-responsive liquid metal-loaded polymeric hydrogels for controlled release of doxorubicin. RSC Adv. 2019, 9, 13026–13032

Round 2
Reviewer 2 Report
Comments and Suggestions for Authors
The paper is acceptable now.